# Developmental Trends in Postural Adjustments During Reaching in Early Childhood

**DOI:** 10.3390/s25072251

**Published:** 2025-04-02

**Authors:** Panchao Zhao, Kai Ma, Tianying Wang, Ziqing Liu

**Affiliations:** 1Department of Physical Education, China University of Geosciences (Beijing), Beijing 100083, China; makai@cugb.edu.cn (K.M.);; 2College of P.E. and Sports, Beijing Normal University, Beijing 100875, China

**Keywords:** postural adjustment, development, early childhood, reaching

## Abstract

Reaching is a fundamental motor skill essential for daily living, with over 50% of daily activities involving reaching movements. Understanding the development of postural adjustments made during reaching in early childhood is crucial for motor development. This study investigated the developmental characteristics of postural adjustments made by children aged 3–6 years during reaching tasks. A total of 135 typically developing children from Chinese kindergartens participated in this study. Kinematic and electromyographic (EMG) data were collected using an infrared motion capture system and surface electromyography, respectively. A two-way ANCOVA was performed to examine the effects of age and gender on kinematic and electromyographic parameters. Post hoc analyses revealed that completion time and shoulder angle showed a linear decreasing trend (*p* < 0.05). The variable wrist joint had an increasing trend in the high-touch task, while the elbow joint showed a nonlinear decreasing trend (*p* < 0.05). EMG results regarding Extensor Carpi Ulnaris (ECU) showed a decreasing trend at all phases (*p* < 0.05). The developmental patterns observed suggest that children progressively utilize more distal (wrist) and fewer proximal (elbow and shoulder) joints during reaching, indicating the maturation of motor patterns. However, the mechanisms of anticipation and compensation for children aged 3–6 are not yet fully understood.

## 1. Introduction

Postural control is essential for numerous daily activities, particularly motor patterns, which consist of reaching, sitting, standing, and walking. It is closely associated with motor development [1]. The period between 3 and 6 years old is critical for motor development, in which basic motor skills and postural control complement and encourage each other [2]. Reaching is a crucial life skill since more than 50% of daily skills rely on it [3]. The execution of smooth, coordinated reaching behaviors, particularly for goal-directed reaching tasks, serves as an indicator of well-organized neuromuscular control and efficient central nervous system integration [4].

Reaching initially emerges in early infancy and develops rapidly within four months of birth, but there is considerable variability in motor characteristics. From nine months to twelve years of age, motor characteristics related to reaching gradually develop and stabilize [5]. Postural stability plays a crucial role in reaching, particularly in supporting sitting position, maintaining head balance, and controlling gaze [6]. Currently, research on reaching predominantly concentrates on muscle activation patterns, mainly using EMG to determine children’s muscle activity patterns. Research indicates that there is significant variability in muscle activity in the postural muscles used for reaching in early childhood [7]. The reaching patterns of children under 2 years old cannot synergistically activate specific neck and trunk muscles, and children under 11 years old do not exhibit a predictable muscle activation pattern before reaching [8]. The existing research has certain limitations. Firstly, the age range of the subjects is broad, and the number of children in each age group is relatively small, making such data unrepresentative. Secondly, studies have mainly focused on muscle activation patterns, with less research on kinematics. Thirdly, there is a lack of exploration of gender differences. These gaps highlight the need for more comprehensive studies on upper-limb postural adjustments made during reaching. Furthermore, investigating the motor characteristics of typically developing children carries significant clinical implications, particularly in establishing normative data for disease screening, informing rehabilitation protocol development, and evaluating therapeutic interventions [9].

The objective of this study was to investigate the developmental features of postural adjustment in early childhood while performing reaching. This study addresses two main issues: (1) What are the motor characteristics of the upper-limb joints during reaching, including in relation to gender differences and age trends? (2) What are the characteristics of anticipation and compensatory postural adjustment among children when reaching?

## 2. Materials and Methods

### 2.1. Participants

This study was conducted at two public kindergartens in Beijing, using a random-sampling method to select participants. The selection process is shown in Figure 1. The testing period extended from May to July 2020. G*power 3.1 software was used to calculate statistics, and the sample size meets the statistical power requirements.

The inclusion criteria were (1) being 3–6 years old; (2) having normal cognitive function and good comprehension skills; and (3) having normal motor ability and the capacity to independently complete sitting and reaching tasks.

The exclusion criteria were (1) physical developmental and skeletal muscle coordination disorders; (2) disorders of cognitive impairment; (3) and an inability to complete the actions required for this study.

The data collection personnel were master’s and doctoral students in relevant fields. They received measurement and practical training before carrying out collection. All data were collected objectively and in a standard format adhering to the guidelines, and incomplete data were deleted.

Through a parent conference, we provided informed consent forms to parents, who voluntarily participated in this project. Written informed consent was obtained from the parents of participants before data collection. This study followed the guidelines of the Declaration of Helsinki, and the Ethics Committee of the Psychology Department at Beijing Normal University approved the protocol (No. 201910210061).

### 2.2. Equipment

Kinematic data were collected using a Bioengineering Technology and Systems (BTS) motion capture system (SMART DX 700, BTS company, Milano, Italy) that sampled at 100 Hz and was used for action recognition. This system’s resolution is 1.5 million pixels. Markers were applied according to the upper-limb model, which is shown in Figure 2.

BTS FREE EMG 300 surface-testing system (BTS company, Milano, Italy) was utilized to obtain EMG data at an acquisition frequency of 1000 Hz. The electrodes’ dimensions were 14.00 mm × 41.50 mm × 24.80 mm, and the maximum transmission distance was 50 m. EMG electrodes were secured to the muscle bellies of the Flexor Carpi Radialis (FCR) and Extensor Carpi Ulnaris (ECU), with a 2 cm distance between the two pieces of the electrode sheet.

### 2.3. Procedures

The test was conducted in an empty classroom in a kindergarten. A small table and chair appropriate for children were situated at the center of the experimental area, surrounded by eight infrared cameras for motion analysis. Afterwards, the participants wore tight-fitting test clothing and affixed reflective markers.

Each subject sat at the front of the chair and placed an object at a forearm’s distance from the edge of the table. Subjects performed two reaching tasks: (1) To simulate eating behavior [10], the target object was placed on the table’s surface; subsequently, the subject reached out to touch the target object, then touched their mouth, and finally put their hand back. (2) To evaluate hand–eye coordination, the object was placed at the subject’s eye level [8], and the subjects were instructed to return their hand after touching the object. Subjects were asked to touch the object as quickly as possible with their dominant hand, defined as the preferred hand for writing, drawing, and eating, and then return this hand to its original position. Each task was tested twice, and the date of the second test was selected for analysis. After the test, the surface of the EMG electrode that was in contact with the skin was wiped with an alcohol swab and sterilized.

### 2.4. Data Processing

The kinematic parameters tested in this study include completion time, shoulder angle, elbow angle, wrist angle, shoulder angular velocity, elbow angular velocity, and wrist angular velocity when the target was touched.

“0” indicates start time, “1” indicates the end, and “0–1” indicates the total phase. “−200–50 ms” indicates the APA phase, and “50–300 ms” indicates the CPA phase [11]. The integrated EMG data of two muscles in the total phase, APA phase, and CPA phase were calculated.

### 2.5. Statistics

The results of the descriptive analyses are presented as means ± standard deviation ($\bar{x}$ ± s). The normal distribution test was performed before statistical analysis; the extreme and singular values were deleted. A two-way ANCOVA was performed to assess the effects of age and gender on kinematic and electromyographic data. The main effect, interaction effect, and results of post hoc analyses were compared via Bonferroni analysis. Data were analyzed with SPSS software, version 23.

## 3. Results

A total of 135 typically developing children aged 3–6 years (mean age = 4.88 ± 0.86 years) who successfully completed all the kinematic and surface electromyography (EMG) assessments were analyzed. The mean height was “111.6 ± 67.65 (cm)”, the mean weight was “20.18 ± 4.28 (kg)”, and the mean BMI was “16.04 ± 1.83 (kg/m^2^)”. Basic information about the participants is shown in Table 1.

Age at completion time and shoulder angle had significant main effects in two tasks (*p* < 0.05). Wrist and elbow angles were associated with a main effect of age in the high-touch tasks (*p* < 0.05). The ECU showed a difference between the main effect of age in total phase and APA phase in the high-touch tasks (*p* < 0.05). The main effect of age of compensation phase appeared in both tasks (*p* < 0.05) (Table 2 and Table 3).

Post hoc analyses revealed significant differences between all the age groups. Completion time and shoulder joint angle showed a linear decreasing trend in two tasks (*p* < 0.05). Wrist joint showed an increasing trend in the high-touch task, while the elbow joint showed a nonlinear decreasing trend (*p* < 0.05). The ECU value showed a decreasing trend in all phases (*p* < 0.05) (Table 4).

## 4. Discussion

As the foundation of fine motor skills, upper-limb function is of great importance in daily human life. This function encompasses actions such as brushing teeth, eating, combing hair, and fastening buttons, all of which significantly influence children’s independence and development during the preschool years [12]. In this study, sport biomechanics analysis was utilized to examine postural adjustment characteristics during reaching tasks carried out by children aged 3–6. There was no significant difference between boys and girls statistically; the age difference was more significant.

Our kinematic analysis incorporated temporal and spatial features. Completion time represents the duration required to execute a motion, providing an intuitive measure of task completion. If additional information is processed during movement, reaction time will increase [13]. In both tasks, completion time had an age-based discrepancy. With an increase in age, there was a decrease in completion time, and the children’s movements became more efficient. This study evaluates the spatial parameters of upper-limb joint angle and angular velocity characteristics. Age differences regarding the wrists and elbows were only noticeable during the high-touch task, whereas there were no differences in the planar-touch tasks. This may be because the high-touch task emphasizes the spatial concept of height, which is more demanding for children, leading to more joint angle differences. The shoulder exhibited age differences in both tasks.

The post hoc analysis demonstrated a linear decrease in shoulder angle, a linear increase in wrist angle, and a linear decrease in elbow angle. Overall, as age increased, more wrist angles and fewer elbow and shoulder angles were used to accomplish the task. The children integrated more distal-limb joint movements in conjunction to complete the tasks, making task completion more precise. This development pattern conforms to the “near-far” principle of motor development in early childhood, which is to develop the proximal-limb joints first and then the distal-limb joints in a sequential manner, in line with the laws of growth and development [14].

Notably, the wrist and shoulder angles displayed linear trends, whereas the elbow angles exhibited a nonlinear pattern of development. Additionally, the kinematic parameters of the 6-year-old participants showed a development trend that differed from that of the other three age groups. This deviating trend may be attributed to the following factors. Between the ages of 4 to 6, children undergo a significant growth period [15]. During this time, the body experiences a period of transition characterized by instability and change as its size changes. These changes can affect joint activity and the development of postural control.

EMG parameters indicate the muscle adjustment strategy adopted during the completion of a movement. The central nervous system utilizes two postural adjustments to achieve movement: anticipatory postural regulation (APA) and compensatory postural regulation (CPA). APA is driven by feedforward control and anticipates movement occurrences in advance, reducing the adverse effects of possible disturbances on postural balance [16]. An anticipation of weight during object lifting is exhibited by children between the ages of 2 to 4 [17], and this ability is available from the first year of life [18]. CPAs are regulated through feedback control mechanisms, and they are integral to movement execution. These adjustments require precise coordination between postural muscle activation and motor strategies to maintain balance and stability during and after movement completion [19]. The compensatory relationship between APAs and CPAs is more apparent in response to disrupted tasks [20]. In comparison to CPAs, APAs are more significant in the prompt identification and diagnosis of children with brain development disorders, developmental coordination disorder, and cerebral palsy [21]. A pair of antagonistic muscles were selected in this study, with the ECU serving as the principal active muscle of the forearm and the FCR acting as the passive muscle for reaching maneuvers. Electromyographic analysis revealed significant age-related differences exclusively in regard to the extensor (ECU), with integrated EMG demonstrating consistent decreasing trends across all movement phases, including both APAs and CPAs, with age. The study findings indicate that children aged 3–6 exhibit APAs during reaching tasks and both control modes follow a consistent pattern. The possible explanations are as follows: First, the children in this age group lack a mature mechanism for the compensation of feedforward and feedback. Second, the two tasks in this study may not be challenging enough for children. Third, the absence of environmental perturbations or external interference likely reduced the compensatory interaction between APAs and CPAs. However, another study on the development of postural adjustments during reaching found that there was no anticipated adjustment in children aged 2–11 [8]. The reason for the difference in the results may be the use of different reaching tasks and differences in the sample sizes of the subjects.

This study offers advantages, such as high reliability and validity of the measurement tools and a focus on upper-limb regulation processes rather than outcomes. Synchronizing the process and outcome evaluations would be intriguing. Nonetheless, there are some limitations to this study, such as the lack of testing for the postural muscles, as we only addressed the characteristics of forearm muscle adjustment. Spinal muscles are crucial for maintaining postural stability during reaching tasks. Collecting simultaneous data from both spinal and leg muscles in future studies can help reveal the role of postural muscles. Additionally, incorporating data from children above 6 years old as well as adult data can allow researchers to better determine the maturity point and facilitate a more comprehensive discussion of age-related differences in reaching tasks.

## 5. Conclusions

In conclusion, age differences exist among children aged 3–6 years, while developmental characteristics remain consistent between boys and girls. The developmental trajectory reveals a distinct shift in movement strategies with an increasing age: children progressively utilize their wrists more while reducing their reliance on elbow and shoulder movements during task execution. This maturation pattern reflects enhanced distal joint control and decreased proximal joint involvement, ultimately resulting in more refined and efficient reaching patterns corresponding to advanced motor development. The EMG parameters indicate more detailed adjustment patterns in reaching movements. At this stage of development, children have not fully developed a compensatory mechanism for feedforward and feedback. Further research is necessary to determine the exact point of maturity for this adjustment mechanism.

## Figures and Tables

**Figure 1 sensors-25-02251-f001:**
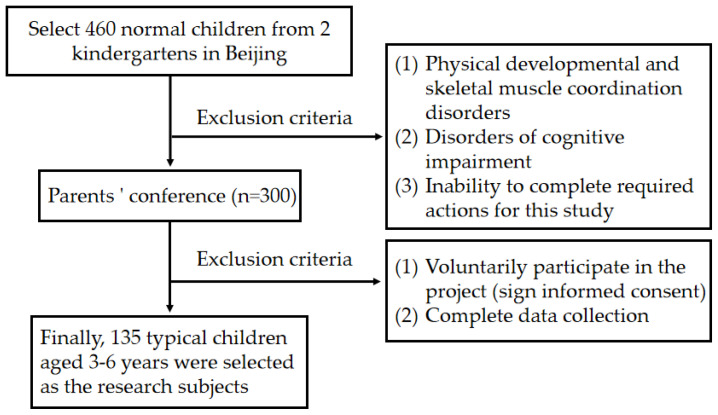
Selection criteria and flowchart for experimental subjects.

**Figure 2 sensors-25-02251-f002:**
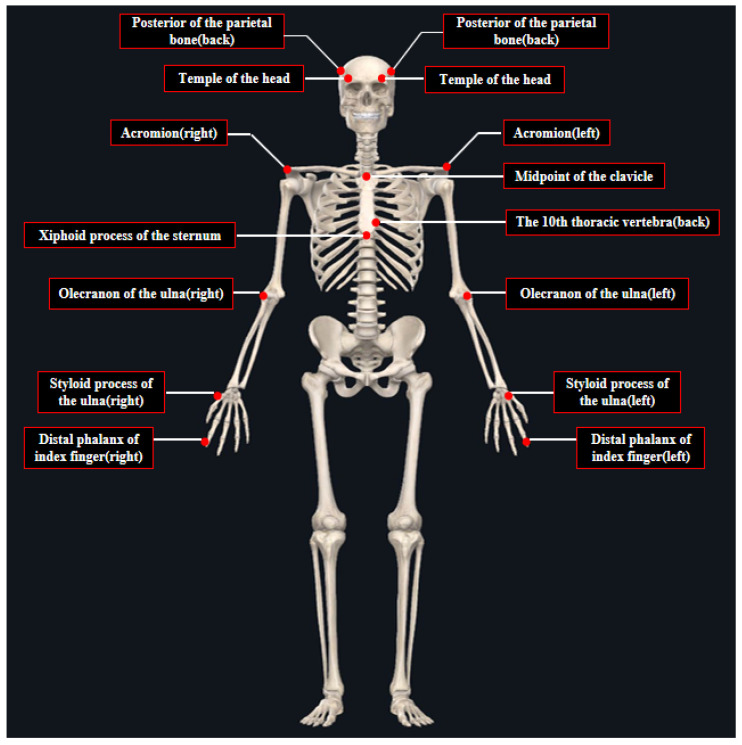
Upper-limb model pertaining to children.

**Table 1 sensors-25-02251-t001:** Basis information on the participants.

Age Group	Age (Years)	Height (cm)	Weight (kg)	BMI (kg/m^2^)
3 years old(*n* = 21)	3.53 ± 0.30	101.15 ± 3.94	16.43 ± 2.54	16.03 ± 2.01
4 years old (*n* = 49)	4.46 ± 0.23	108.52 ± 3.98 a	18.66 ± 2.39	15.79 ± 1.22
5 years old (*n* = 49)	5.45 ± 0.23	116.13 ± 5.08 ab	22.14 ± 4.61 ab	16.28 ± 2.24
6 years old (*n* = 16)	6.22 ± 0.13	121.36 ± 4.01 abc	23.73 ± 3.96 ab	16.03 ± 1.78
All (*n* = 135)	4.88 ± 0.86	111.6 ± 67.65	20.18 ± 4.28	16.04 ± 1.83

Notes: “a” means a significant difference compared with the 3-year-old group (*p* < 0.05), “b” means a significant difference compared with the 4-year-old group (*p* < 0.05), and “c” means a significant difference compared with the 5-year-old group (*p* < 0.05).

**Table 2 sensors-25-02251-t002:** Main and interaction effects of kinematic parameters.

Dependent Variable	Tasks	Effect Level	df	F	sig	η^2^
Completion Time (s)	Planar touch	Age	3	21.263	0.000	0.347
		Gender	1	0.169	0.681	0.001
	High touch	Age	3	5.500	0.001	0.124
		Gender	1	0.000	0.987	0.000
Wrist Angle (deg)	Planar touch	Age	3	1.205	0.311	0.029
		Gender	1	0.112	0.739	0.001
	High touch	Age	3	2.808	0.043	0.067
		Gender	1	2.648	0.106	0.022
Elbow Angle (deg)	Planar touch	Age	3	1.823	0.147	0.043
		Gender	1	1.11	0.294	0.009
	High touch	Age	3	4.607	0.004	0.105
		Gender	1	1.043	0.309	0.009
Shoulder Angle (deg)	Planar touch	Age	3	9.885	0.000	0.197
		Gender	1	1.89	0.172	0.015
	High touch	Age	3	7.939	0.000	0.168
		Gender	1	0.072	0.788	0.001
Wrist Angular Velocity (rad/s)	Planar touch	Age	3	1.045	0.375	0.025
		Gender	1	0.008	0.929	0.000
	High touch	Age	3	0.489	0.690	0.012
		Gender	1	0.430	0.513	0.004
Elbow Angular Velocity (rad/s)	Planar touch	Age	3	1.758	0.159	0.042
		Gender	1	4.032	0.047	0.032
	High touch	Age	3	0.795	0.499	0.020
		Gender	1	0.023	0.879	0.000
Shoulder Angular Velocity (rad/s)	Planar touch	Age	3	1.029	0.382	0.025
		Gender	1	0.639	0.426	0.005
	High touch	Age	3	0.878	0.455	0.022
		Gender	1	0.056	0.813	0.000

Notes: The effect level includes the main effect and the interaction effect, and “sig” represents the significant difference in effect level.

**Table 3 sensors-25-02251-t003:** Main and interaction effects of EMG parameter.

Dependent Variable	Tasks	Effect Level	df	F	sig	η^2^
ECU (0–1) (mV·s)	Planar touch	Age	3	2.339	0.077	0.055
		Gender	1	0.020	0.888	0.000
	High touch	Age	3	5.004	0.003	0.113
		Gender	1	0.072	0.789	0.001
ECU (APAs) (mV·s)	Planar touch	Age	3	1.718	0.167	0.041
		Gender	1	0.001	0.974	0.000
	High touch	Age	3	6.646	0.000	0.145
		Gender	1	0.081	0.777	0.001
ECU (CPAs) (mV·s)	Planar touch	Age	3	5.312	0.002	0.116
		Gender	1	0.361	0.549	0.003
	High touch	Age	3	7.914	0.000	0.168
		Gender	1	0.451	0.503	0.004
FCR (0–1) (mV·s)	Planar touch	Age	3	1.456	0.230	0.035
		Gender	1	1.000	0.319	0.008
	High touch	Age	3	0.332	0.802	0.008
		Gender	1	0.142	0.707	0.001
FCR (APAs) (mV·s)	Planar touch	Age	3	0.521	0.669	0.013
		Gender	1	4.596	0.050	0.037
	High touch	Age	3	0.400	0.753	0.010
		Gender	1	2.792	0.097	0.023
FCR (CPAs) (mV·s)	Planar touch	Age	3	2.372	0.074	0.056
		Gender	1	0.051	0.822	0.000
	High touch	Age	3	3.125	0.050	0.074
		Gender	1	0.002	0.969	0.000

Notes: The effect level includes the main effect and the interaction effect, and “sig” represents the significant difference in effect level.

**Table 4 sensors-25-02251-t004:** Multiple-comparisons ANOVA of kinematic parameters.

Dependent Variable	Tasks	3 (*n* = 21)	4 (*n* = 49)	5 (*n* = 49)	6 (*n* = 16)	All (*n* = 135)
Completion Time (s)	Planar touch	7.03 ± 1.52	5.12 ± 1.19 a	4.92 ± 1.09 a	4.13 ± 0.35 abc	5.23 ± 1.41
	High touch	4.44 ± 1.02	3.53 ± 0.89 a	3.46 ± 0.74 a	3.38 ± 1.47 a	3.61 ± 1.00
Wrist Angle (deg)	Planar touch	153.86 ± 18.07	161.11 ± 26.53	163.63 ± 11.93	165.67 ± 7.95	161.43 ± 19.40
	High touch	149.31 ± 20.86	155.91 ± 12.29	157.94 ± 12.46	162.04 ± 11.35 a	156.42 ± 14.13
Elbow Angle (deg)	Planar touch	151.96 ± 12.48	147.55 ± 14.90	150.15 ± 13.63	155.42 ± 10.64	150.12 ± 13.72
	High touch	145.46 ± 18.30	137.55 ± 15.61	130.60 ± 17.04 a	141.27 ± 14.25	136.73 ± 17.04
Shoulder Angle (deg)	Planar touch	128.83 ± 8.99	121.38 ± 12.04	115.97 ± 11.67 a	111.50 ± 9.07 ab	119.42 ± 12.20
	High touch	148.79 ± 14.88	143.59 ± 18.13	130.89 ± 15.86 ab	129.59 ± 15.03 ab	138.06 ± 17.95
ECU (0–1) (mV·s)	Planar touch	0.03 ± 0.01	0.03 ± 0.01	0.03 ± 0.01	0.03 ± 0.01	0.03 ± 0.01
	High touch	0.04 ± 0.02	0.03 ± 0.02 a	0.03 ± 0.01 a	0.02 ± 0.01 a	0.03 ± 0.02
ECU (APAs) (mV·s)	Planar touch	0.02 ± 0.02	0.02 ± 0.02	0.02 ± 0.01	0.01 ± 0.00	0.02 ± 0.01
	High touch	0.03 ± 0.02	0.02 ± 0.02 a	0.02 ± 0.01 a	0.01 ± 0.01 a	0.02 ± 0.01
ECU (CPAs) (mV·s)	Planar touch	0.04 ± 0.02	0.02 ± 0.02	0.02 ± 0.01 a	0.02 ± 0.01 a	0.02 ± 0.02
	High touch	0.04 ± 0.02	0.03 ± 0.02 a	0.02 ± 0.01 a	0.01 ± 0.01 a	0.03 ± 0.02

Notes: “a” means a significant difference compared with the 3-year-old group (*p* < 0.05). “b” means a significant difference compared with the 4-year-old group (*p* < 0.05). “c” means a significant difference compared with the 5-year-old group (*p* < 0.05).

## Data Availability

The original contributions presented in this study are included in the article; further inquiries can be directed to the corresponding author.

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
