# Peer review of "Developmental Trends in Postural Adjustments During Reaching in Early Childhood"

_sensors, 2025, doi:10.3390/s25072251_

Round 1
Reviewer 1 Report
Comments and Suggestions for Authors
The study consider characteristics of postural adjustment in children aged 3-6 years old during reaching and authors mainly concentrated at the motion age-gender dependency calculating. The paper well written and clear to understand, however several issues should be improved before acceptence:
For the section 2.1 graphical flowchart is needed
For the section 2.2 photograph or sketch how markers were applied to the subjects, acronim EMG should be described earlier, more detailed explanation of the equipment shold be given: resolution of the detection system, error of data assimilation etc.
For the section 2.3. line 93 - what kind of ask? visual-audial-another kind of signal?
Results and discussion: Table 5 better be represented graphically (or supported graphically), it is interesting to compare thre results obtained with the results from [10,7]. According to my previos remark - is there any diffirence in what kind of command for reaching was used (please discuss it- data\opinion, in any way)?
Author Response
-----------------------------------------------------------------------------------------
Reply to the Review Report of Sensors-3376460-Panchao Zhao
-----------------------------------------------------------------------------------------
We are very grateful to the Review for the helpful comments and useful suggestions on our manuscript Sensors-3376460. We have read this review carefully and have revised our manuscript according to the review’s comments. Revised or/and added parts are marked with the red text in the manuscript. The follows are our reply item by item:
1、Comment: For the section 2.1 graphical flowchart is needed.
Reply: Thanks for this reminding and helpful suggestion. The graphical flowchart has been placed in section 2.1.
2、Comment: For the section 2.2 photograph or sketch how markers were applied to the subjects, acronim EMG should be described earlier, more detailed explanation of the equipment should be given: resolution of the detection system, error of data assimilation etc.
Reply: Thanks for this reminding and helpful suggestion. The upper-limb model of markers has been added in 2.2, and the EMG paste position is described at earlier position. The resolution of the BTS system is 1.5 million pixels, and the added content has been highlighted in red at the corresponding position in the text.
3、Comment: For the section 2.3. line 93 - what kind of ask? visual-audial-another kind of signal?
Reply: This is not a question, but a starting instruction to finish the task. Visual and auditory cues are not signals.
4、Comment: Results and discussion: Table 5 better be represented graphically (or supported graphically), it is interesting to compare three results obtained with the results from [10,7]. According to my previous remark - is there any difference in what kind of command for reaching was used (please discuss it- data\opinion, in any way)
Reply: Thanks for this reminding and helpful suggestion. I tried to use a graph to represent this part of the data, as shown in the figure in DOC below. I felt that the effect was not as good as a table, so I kept the form of the table. The two commands focus on two aspects. The first is to simulate eating tasks, and the second examines children's hand-eye coordination ability. The comparison of the two task is presented at the end of the second paragraph, and the comparison with previous research is presented at the end of the fifth paragraph, both of which are highlighted in red in the text.
Thank you again for your time in reviewing our manuscript.
Best wishes for you.
Sincerely yours,
Panchao Zhao, Ka Ma, Tianying Wang, Ziqing Liu.
March 9, 2025.

Reviewer 2 Report
Comments and Suggestions for Authors
Thank you for allowing me to review this manuscript. The manuscript investigates the developmental achievement of key kinematic data measures of movement tasks involved with reaching and correlates attainment of those tasks with age and gender of children 3 to 6 years of age.
While I think the authors reach valid conclusions, the manuscript needs some reorganization to present the research in a more logical format. The manuscript would benefit from overall English editing. In my opinion, the title of the manuscript as it is currently, gives the reader an impression of descriptive analysis, rather than anything further. The authors might consider a change to something along the lines of “Postural adjustment development trends of reaching in children aged 3-6 years old.” The following are my constructive comments to the authors intended to strengthen the overall manuscript.
Introduction
Clearly state the gap in current knowledge. Lines 37-39 sum up current research. The following line is “Further research is required, particularly on children with disorders.” What research is needed? Tell what we know; then - however, we do not know………. Therefore, we need further research to investigate ……… Be precise with what is needed. I don’t think the authors have fully justified the importance of why this study was needed. Was it to establish normal values for reaching measures at each developmental level? And in doing so, then we are able to identify when children do not meet the “norms?” I am unsure why the discussion about children with disorders and children with illnesses etc., in lines 40-44 exists. It seems that these children were excluded from participation in the study (lines 54-59).
Materials and Methods
Please give the recruitment goal. Did the sample number provide statistical power?
Line 60, number 1: This is not necessary. This is in your exclusion criteria above.
Line 61, number 2: Participants that request to withdraw from the study are not excluded, they are withdrawn. There should be a statement a to what will be done with the data already collected from a participant that wishes to withdraw.
Line 62, number 3: The handling of this data should be described in the data analysis section. This is considered handling of “missing data.” The participant is not excluded, but you may choose to exclude the data.
Please describe how recruitment and sampling occurred beyond “random sampling.” The description of the process should allow the process to be reproducible.
Please describe the informed consent process prior to telling about the testing process.
Were parents present for the testing?
Procedures, lines 89-92: Task 1 lists the purpose of the task as “to simulate the eating behavior,” however, I do not clearly see a purpose for task 2.
Date (should be data) processing, lines 102-105: The text is confusing when placed here. I’m not sure it is needed.
Table notes
Please put Table notes on the appropriate Tables.
The Tables could be presented more concisely and would be less confusing.
Discussion
The first paragraph in the discussion section is template text and needs to be removed.
Lines 147-149: To which developmental characteristics are you referring? Do you mean there is consistency with previous research? Previously identified norms?
Line 166: Please give a brief description of the principle of “near to far.” Is this consistent with previous research?
Line 205: “Including data from adults can provide a more comprehensive discussion…..” What about the ages between 6 years of age and adulthood? Would this be needed to identify the point of maturity?
Comments on the Quality of English LanguageThe manuscript would benefit from English editing.
Author Response
-----------------------------------------------------------------------------------------
Reply to the Review Report of Sensors-3376460-Panchao Zhao
-----------------------------------------------------------------------------------------
We are very grateful to the Review for the helpful comments and useful suggestions on our manuscript Sensors-3376460. We have read this review carefully and have revised our manuscript according to the review’s comments. Revised or/and added parts are marked with the red text in the manuscript. The follows are our reply item by item:
Introduction
1、Comment: Clearly state the gap in current knowledge. Lines 37-39 sum up current research. The following line is “Further research is required, particularly on children with disorders.” What research is needed? Tell what we know; then - however, we do not know………. Therefore, we need further research to investigate ……… Be precise with what is needed. I don’t think the authors have fully justified the importance of why this study was needed. Was it to establish normal values for reaching measures at each developmental level? And in doing so, then we are able to identify when children do not meet the “norms?” I am unsure why the discussion about children with disorders and children with illnesses etc., in lines 40-44 exists. It seems that these children were excluded from participation in the study (lines 54-59).
Reply: Thanks for this reminding and helpful suggestion. I have revised the description of children with illnesses in the article and added content to elaborate on the current research status and the importance of this article, which has been highlighted in red in the text.
Materials and Methods
2、Comment: Please give the recruitment goal. Did the sample number provide statistical power?
Reply: Thanks for this reminding and helpful suggestion. The statistical data was calculated using G power software, with 20 children in each group and a total of 80 people, which can meet the statistical power. This article has a total of 135 children. In the text, it changed in the text: “Using G power software to calculate statistics, the sample size meets the statistical power requirements”.
3、Comment: Line 60, number 1: This is not necessary. This is in your exclusion criteria above.
Reply: Thanks for this reminding and helpful suggestion. The “Shedding and rejection criteria” section has been completely removed, while the “exclusion criteria” section remains.
4、Comment: Line 61, number 2: Participants that request to withdraw from the study are not excluded, they are withdrawn. There should be a statement to what will be done with the data already collected from a participant that wishes to withdraw.
Reply: Thanks for this reminding and helpful suggestion. This sentence is ambiguous, and the entire part has been deleted.
5、Comment: Line 62, number 3: The handling of this data should be described in the data analysis section. This is considered handling of “missing data.” The participant is not excluded, but you may choose to exclude the data.
Reply: Thanks for this reminding and helpful suggestion. The sentence has been deleted, please refer to the “Participants” section in the text.
6、Comment: Please describe how recruitment and sampling occurred beyond “random sampling.” The description of the process should allow the process to be reproducible.
Reply: Thanks for this reminding and helpful suggestion. I created a participant recruitment process flowchart in section 2.1, which provided a detailed introduction to the recruitment process.
7、Comment: Please describe the informed consent process prior to tell about the testing process.
Reply: Thanks for this reminding and helpful suggestion. Through a parent conference, we provided informed consent to parents and voluntarily participated in this project. Details can be found in the fifth paragraph of the “Participant” section, and relevant content has been added to the article.
8、Comment: Were parents present for the testing?
Reply: Parents did not participate in the test, only signed an informed consent.
9、Comment: Procedures, lines 89-92: Task 1 lists the purpose of the task as “to simulate the eating behavior,” however, I do not clearly see a purpose for task 2.
Reply: Task 2 observes ability of hand-eye coordination and whether it can accurately touch the target point after instructions are given. However, this article does not have statistical accuracy indicators, only observes process indicators, which can be supplemented in future research. I have added content in the corresponding section of the text.
10、Comment: Date (should be data) processing, lines 102-105: The text is confusing when placed here. I’m not sure it is needed.
Reply: Thanks for this reminding and helpful suggestion. I have made modifications in the text.
Table notes
11、Comment: Please put Table notes on the appropriate Tables.
Reply: Thanks for this reminding and helpful suggestion. Notes have been added below each table, and two notes have been added, highlighted in red below the tables in the text.
12、Comment: The Tables could be presented more concisely and would be less confusing.
Reply: Thanks for this reminding and helpful suggestion. The interaction effects in Table 2-3 did not show statistical differences, so they were removed to make the table appear more concise.
Discussion
13、Comment: The first paragraph in the discussion section is template text and needs to be removed.
Reply: I'm so sorry, this paragraph has been deleted.
14、Comment: Lines 147-149: To which developmental characteristics are you referring? Do you mean there is consistency with previous research? Previously identified norms?
Reply: Thanks for this reminding and helpful suggestion. The meaning here is that the developmental characteristics of boys and girls are the same, because there is no significant difference in the data between them statistically. My statement may be a bit problematic, and I have made the necessary modifications and highlighted them in red in the text.
15、Comment: Line 166: Please give a brief description of the principle of “near-to- far.” Is this consistent with previous research?
Reply: Thanks for this reminding and helpful suggestion. The “near-far” principle of early motor development in children refers to the gradual development from the central part of the body to the peripheral parts, with the shoulder joint approaching the center of the body and developing first, and the wrist joint moving away from the body and developing later. The research results of this article comply with the “near-far” principle, and I have added and modified relevant content and highlighted it in red in the article.
16、Comment: Line 205: “Including data from adults can provide a more comprehensive discussion…” What about the ages between 6 years of age and adulthood? Would this be needed to identify the point of maturity?
Reply: Thanks for this reminding and helpful suggestion. The data of children above 6 years old can determine the maturity point, which has been modified in the corresponding section of the article and highlighted in red.
17、Comment: The manuscript would benefit from English editing.
Reply: Thanks for this reminding and helpful suggestion. I have polished the entire text and highlighted the modified parts in red.
Thank you again for your time in reviewing our manuscript.
Best wishes for you.
Sincerely yours,
Panchao Zhao, Ka Ma, Tianying Wang, Ziqing Liu.
March 9, 2025.
